PCAtest: testing the statistical significance of Principal Component Analysis in R

http://orcid.org/0000-0001-7734-8679 Camargo Arley arley.camargo@gmail.com
Centro Universitario Regional Noreste, Universidad de la República , Rivera , Uruguay
Wilke Claus
Electronic publication date: 2022 Feb 17
Publication date: 2022
Volume: 10
Electronic Location ID: e12967
Received 2021 Aug 19; Accepted 2022 Jan 28
Copyright: © 2022 Camargo
Copyright year: 2022
Copyright holder: Camargo
License: This is an open access article distributed under the terms of the Creative Commons Attribution License, which permits unrestricted use, distribution, reproduction and adaptation in any medium and for any purpose provided that it is properly attributed. For attribution, the original author(s), title, publication source (PeerJ) and either DOI or URL of the article must be cited.
License URL: https://creativecommons.org/licenses/by/4.0/

Keywords: Principal component analysis, Statistical significance, Permutation, R function, PCAtest

Funding: Programa de Desarrollo de las Ciencias Básicas Sistema Nacional de Investigadores, Agencia Nacional de Investigación e Innovación This work was supported by the Programa de Desarrollo de las Ciencias Básicas (PEDECIBA, Uruguay) and the Sistema Nacional de Investigadores, Agencia Nacional de Investigación e Innovación (SNI-ANII, Uruguay). The funders had no role in study design, data collection and analysis, decision to publish, or preparation of the manuscript.

==============================
Principal Component Analysis (PCA) is one of the most broadly used statistical methods for the ordination and dimensionality-reduction of multivariate datasets across many scientific disciplines. Trivial PCs can be estimated from data sets without any correlational structure among the original variables, and traditional criteria for selecting non-trivial PC axes are difficult to implement, partially subjective or based on ad hoc thresholds. PCAtest is an R package that implements permutation-based statistical tests to evaluate the overall significance of a PCA, the significance of each PC axis, and of contributions of each observed variable to the significant axes. Based on simulation and empirical results, I encourage R users to routinely apply PCAtest to test the significance of their PCA before proceeding with the direct interpretation of PC axes and/or the utilization of PC scores in subsequent evolutionary and ecological analyses.

Introduction

Principal Component Analysis (PCA), originally developed by Pearson (1901) and Hotelling (1933), is one of the most broadly used statistical methods for the ordination and dimensionality-reduction of multivariate datasets across many scientific disciplines with many adaptations for different goals and data types (Manly, 1986; Jackson, 1991; Vieira, 2012; Jolliffe & Cadima, 2016). PCA estimates independent, linear combinations of the original (possibly correlated) variables in such a manner that the first component (PC1) accounts for the largest possible variance in the data, and each subsequent PC accounts for decreasing proportions. In most applications, PC scores (also called z-scores or l-scores; see Jackson, 1991) are calculated for the observations based on those PC axes considered to adequately account for the correlational structure in the data. These PC scores are used to plot and find groupings of the observations in bi- or tri-dimensional spaces defined by the orthogonal PC axes (Manly, 1986; Jackson, 1991) and also to look for potential associations with explanatory variables (e.g., correlation analysis between PC scores obtained from morphological variables and environmental variables of the same observations). In addition, the loading matrix is usually inspected to find out which of the original variables contributed the most to each of the relevant PCs (Manly, 1986; Jackson, 1991; Vieira, 2012).

It is well known that random (or trivial) PCs can be estimated from data sets without any correlational structure among the original variables due to sampling error, especially when number of observations is low relative to the number of variables (Manly, 1986; Jackson, 1993; Vieira, 2012). Even when there is correlation among the original variables, it is necessary to retain the PC axes that summarize that real structure for downstream inferences, and discard the PCs that do not account for more variation than that expected by chance due to sampling error, which creates spurious correlation among the original variables. The criteria traditionally applied for selecting the number of PC axes to be retained in subsequent analyses are difficult to implement, partially subjective or based on ad hoc thresholds (Vieira, 2012). For instance, a traditional ‘rule-of-thumb’ criterion considers that PC axes with eigenvalues larger than one should be retained because they explain more variation than that of a single original variable. Another criterion found in the literature suggests that one should retain as many PC axes as necessary to account for most of the variation in the data, which usually leads to the retention of random axes that explain small portions of variance. Finally, a popular approach consists in visually inspecting a scree plot to find an “elbow” in the distribution of observed eigenvalues that represents the point at which eigenvalues become small and level off as expected from random noise. In general, when following these criteria, it becomes very difficult to ascertain if a given PC axis should be retained when its eigenvalue is very close to that one expected by random chance (Vieira, 2012). Because these ad hoc procedures are strongly dependent on distributional assumptions, permutation-based statistical tests are ideal for evaluating the significance of PCA (Manly, 1986; Jackson, 1993; Vieira, 2012). Empirical evidence and simulation studies have shown the permutation-based tests (or more generally, ‘parallel analysis’) are the most consistently accurate under a range of conditions (Manly, 1986; ter Braak, 1988; Dijksterhuis & Heiser, 1995; Peres-Neto, Jackson & Somers, 2005). Therefore, permutational methods are ideal when one needs to select those PCs representing systematic sources of variation in multivariate data and discard other PCs that only reflect random noise caused by sampling error.

A permutation-based test consists in shuffling individual values within each observed variable to break the original correlation structure among variables without affecting their distributions, calculating the test statistic for each permutation to build a null distribution, and to compare this distribution with the observed statistics (ter Braak, 1988; Dijksterhuis & Heiser, 1995; Peres-Neto, Jackson & Somers, 2005; Vieira, 2012; Vitale et al., 2017). Consequently, the null hypothesis that the variables used in the PCA are uncorrelated with each other can be tested with null distributions of test statistics generated via data permutation. Vieira (2012) proposed to evaluate the significance of PCA with two test statistics that summarizes variation in eigenvalues alone (ψ) or in combination with the number of variables (φ; Gleason & Staelin, 1975). If these statistics show that a PCA is meaningful, then null distributions can also be used to select the number of significant PC axes using for instance the rank-of-roots statistic (ter Braak, 1988). Finally, Björklund (2019) also suggested to use Vieira (2012)’s index of the loadings to test the significance of the contributions of the original variables (eigenvector loadings) to each significant PC also via permutation. Vieira (2012) implemented these tests of PCA significance in the proprietary commercial MATLAB program (typically used by mathematicians and engineers), but an equivalent application in open-source free R software (R Core Team, 2021), more broadly used in life and environmental sciences, is still lacking. While there are R packages implementing ‘parallel analysis’ to select the number of PCs, such as nFactors (Raîche & Magis, 2020), there are no implementations for testing the overall significance of a PCA or the loadings of each significant PC.

Materials and Methods

Herein, I introduce an R package to perform the statistical tests proposed by Vieira (2012) to evaluate the overall significance of a PCA, the significance of each PC axis, and of the variable loadings for the significant axes. This R package is named PCAtest and is available for download from https://github.com/arleyc/PCAtest. Eigenvalues and eigenvector loadings are calculated from a standardized (centered and scaled) multivariate matrix (variables in columns and observations in rows) with the stats:prcomp function, which performs a singular value decomposition of the correlation matrix (Choi & Yang, in press). Observed eigenvalues are used to calculate empirical ψ and φ values, percentages of explained variance for each PC axis (rank-of-roots statistic; ter Braak, 1988), and the indexes of the loadings of each variable on each PC axis (Vieira, 2012). Next, PCAtest performs a bootstrap resampling of the observed data using base::sample to sample rows from the data set with replacement and calculates all statistics for each bootstrap replicate. Bootstrap resampling is a classical and well-established numerical methodology for calculating confidence intervals of statistics based on approximation of their sampling distributions (Efron, 1979; Efron & Tibshirani, 1986; Manly, 1991). To build null distributions for each statistic, PCAtest permutes observations within each variable by sampling without replacement within columns (i.e., applying base::sample to all columns with base::apply), and calculates all statistics for each random permutation. By default, the function runs 1,000 random permutations and bootstrap replicates of the empirical data. Based on the bootstrap resampling and permutation, 95%-confidence intervals around mean values are calculated with stats::quantile. Finally, P-values are calculated as the fraction of null statistics larger than the observed statistics in relation to the number of random permutations. Significant P-values for ψ and φ imply that there is non-random correlational structure in the data, and that a PCA is biologically meaningful. Significant eigenvalues mean that the respective PC axes reflect non-random correlations among variables, and significant loadings mean that the respective variables have a larger contribution in the PC score beyond random noise. The nominal alpha level by default is set to 0.05, but it can be changed by the user to explore the effects of more conservative vs liberal Type I error.

The function creates a list object including the following elements: (1) the empirical ψ and φ values, (2) the ψ and φ values derived from permutation, (3) the percentage of variation explained by each empirical PC, (4) the percentage of variation explained by each PC with the permuted and bootstrapped data, (5) the index loadings of each PC with the permuted and bootstrapped data, and optionally, (6) the correlations of each PC with the original variables using the empirical, the permuted, and the bootstrapped data. Messages are displayed in the console during the run informing the user about the progress of the analysis at each major step: bootstrap replication, random permutation, and statistical tests. After completing these steps, the function displays a summary of the data set dimensions, the randomization results, and the recommendations for subsequent analyses. First, the output describes the size of the dataset and the number of permutations and bootstraps. Next, the randomization results for each statistic and eigenvalue are displayed including the empirical value, the range of randomized values, and the corresponding P-value. Based on these results, the output informs how many PC axes are significant and how much percentage of the total variation they account for (mean and 95%-confidence interval). Finally, based on the randomization of PC loadings, the output lists the variables with significant loadings for each of the retained PC axes. The user can also choose to calculate the correlations of each PC axis with each of the observed variables as an alternative to the index of the loadings, which are prone to high Type I error in large data sets (Vieira, 2012). If the empirical ψ and φ statistics are not significant, the function reports the results of these randomization tests only.

The function produces four types of plots: (1) the null distributions of randomized ψ values and the empirical value, (2) the null distribution of randomized φ values and the empirical value, (3) the percentage of total variation explained by each randomized and empirical PC (mean and 95%-confidence intervals), and (4) the index loadings of the randomized and empirical values of significant PC axes (mean and 95%-confidence intervals). The function creates up to four plots in a single page, with additional plots shown in as many new pages as necessary given the number of significant PCs. Empirical statistics (means and confidence intervals based on bootstrap resampling) are plotted in red color and null statistics based on random permutation (means and confidence intervals) are plotted in gray color. The user can opt to suppress the plots that are produced by default.

Results

To demonstrate the use of the PCAtest, I simulated several datasets with varying levels of correlation structure among the variables and reanalyzed more complex datasets found in the literature. The implementation of these analyses is included in the package and it is also available as a tutorial in the website https://arleyc.github.io/PCAtest/.

I simulated three datasets consisting of five variables with 100 observations and varying levels of correlation among the variables (r = 0, 0.25, and 0.5). These artificial datasets (100 replicates for each value of r) were generated with the R function MASS:mvrnorm specifying a mean equal to zero (mu = 0) and the covariance matrix (Sigma) with variances = 1 and covariances = r for all five normally-distributed variables. In the first case, a PCA should not find any significant correlation structure among the variables. In the second and third examples, the PCA should be significant and PC1 should capture most of the variance because the simulated correlation structure is identical among all variables. In the first example, PCAtest did not detect any significant correlation structure as expected in 95% of the replicates (Fig. 1, Table S1). In the second example, the first PC axis was significant in 100% of the replicates, while the second PC axis was also significant in one replicate (Fig. 2, Table S2). In the third example, the first PC axis was significant in 100% of the replicates (Fig. 3, Table S3).

Figure 1 Null distributions and empirical statistics derived from PCAtest analysis of simulated data consisting of five uncorrelated variables and 100 observations.

Figure 2 Null distributions and empirical statistics derived from PCAtest analysis of simulated data consisting of five correlated variables (r = 0.25) and 100 observations.

Lower plots show mean observed values (red dots), 95%-confidence interval (CI) based on 1,000 bootstrap replicates (red bars), mean values and 95%-CI based on 1,000 random permutations (gray dots and bars, respectively).

Figure 3 Null distributions and empirical statistics derived from PCAtest analysis of simulated data consisting of five correlated variables (r = 0.50) and 100 observations.

Lower plots show mean observed values (red dots), 95%-confidence interval (CI) based on 1,000 bootstrap replicates (red bars), mean values and 95%-CI based on 1,000 random permutations (gray dots and bars, respectively).

I also re-analyzed a published dataset with PCAtest to emphasize the need for testing non-randomness of PCA axes by assessing sampling variance in the empirical data. Recently, Wong & Carmona (2021) used a PCA analysis to reduce dimensionality of a dataset consisting of seven morphological variables of 29 ant species. They retained the first two PC axes based on the selection criterion that eigenvalues greater than one are biologically-meaningful, and then used the PC scores from these axes in ecological analyses of functional diversity. The re-analysis shows that the first two eigenvalues are greater than one (3.85 and 1.52 for PC1 and PC2, respectively), and the significant ψ and φ statistics suggest that a PCA is able to extract non-random correlation structure among the morphological variables (Figs. 4A, 4B; Table S4). However, the randomization test with the ‘rank-of-roots’ statistic (percentage of explained variance) indicates that PC2 is not significant (P-value = 0.16, Fig. 4C). Furthermore, the variable ‘eye width’ does not contribute a significant loading to PC1 (Fig. 4D, variable 6). Consequently, Wong & Carmona (2021) should only have used the PC1 and ignored the contribution of the ‘eye width’ variable to the scores of this PC axis in their ecological analyses. While the inclusion of non-significant PC axes and variables might not have impacted the outcome of their analyses, it probably added additional noise and uncertainty to the interpretation of their results.

Figure 4 Null distributions and empirical statistics derived from PCAtest analysis of seven morphological variables measured in 29 ant species (data from Wong & Carmona (2021)).

Lower plots show mean observed values (red dots), 95%-confidence interval (CI) based on 1,000 bootstrap replicates (red bars), mean values and 95%-CI based on 1,000 random permutations (gray dots and bars, respectively).

In order to exemplify a data set with more variables than observations, I also re-analyzed the microarray data of Ringnér (2008) containing the expression profiles of 8,534 genes screened in 105 samples (Data S1). The author did not evaluate the significance of the PCA, but used the first two PCs to project the samples in a bi-dimensional scatter plot, which together accounted for only 18.7% of the total variation. The analysis with PCAtest using 1,000 random permutations found significant both ψ and φ values (Figs. 5A, 5B) and 21 significant PC axes accounting for 63.4% of the total variation (Fig. 5C). While the index of the loadings found 7,419 variables (i.e., genes) contributing significantly to PC1, only a subset of 2,612 variables had significant correlations with this PC. These results suggest that the index of the loadings falsely finds too many significant results for irrelevant variables in large data sets as Vieira (2012) found with simulated data. Given the complexity of this dataset with many more variables than observations, I transposed the matrix using the function base::t to run a Q-mode PCA analysis sensu Lee, Liong & Jemain (2017), which resulted in six significant PCs accounting for 79% of the total variance (Fig. 6, Table S5). Moreover, PC1 alone explained 67% suggesting a strong correlation structure, meaning a shared gene expression profile among samples, and that PC1 scores from the Q-mode analysis could be more informative for discovering gene groupings than the R-mode PCA used by Ringnér (2008).

Figure 5 Results of R-mode PCAtest analysis of microarray data from Ringnér (2008).

Null distributions and empirical statistics derived from PCAtest analysis of 8,534 genes screened in 105 samples (data from Ringnér, 2008). Lower plots show mean observed values (red dots), 95%-confidence interval (CI) based on 1,000 bootstrap replicates (red bars), mean values and 95%-CI based on 1,000 random permutations (gray dots and bars, respectively).

Figure 6 Results of Q-mode PCAtest analysis of microarray data from Ringnér (2008).

Null distributions and empirical statistics derived from PCAtest analysis of 8,534 genes (observations) screened in 105 samples (variables). The original data set of Ringnér (2008) was transposed to perform a Q-mode PCA analysis. Lower plots show mean observed values (red dots), 95%-confidence interval (CI) based on 1,000 bootstrap replicates (red bars), mean values and 95%-CI based on 1,000 random permutations (gray dots and bars, respectively).

Discussion

The overall significance of a PCA analysis has been poorly addressed in mainstream evolutionary biology journals (Björklund, 2019). However, the lack of a rigorous statistical evaluation of PCA significance is common place across a variety of scientific disciplines as demonstrated in a systematic search of papers published during 2021 (until Nov 30th) in PeerJ journals. Using the keyword ‘PCA’, I found that none of 155 articles evaluated the overall significance of the PCA, and that only 26 (16.8%) used an explicit criterion for PC selection (Table 1, Table S6). Three criteria were applied to select the number of relevant PC axes, namely the “cumulative variance”, “eigenvalue > 1”, and “scree plot” criteria, although they are known to suffer of drawbacks or are difficult to apply in practice. Often, these criteria lead to the consideration of PC axes that only represent random noise in the data, but not correlation structure among the original variables. This is the case with the criterion of including as many PC axes as necessary to reach a pre-specified level of explained variation (Peres-Neto, Jackson & Somers, 2005), which in most cases results in the consideration of trivial, non-significant axes. In the literature review, I found that when this criterion was used (14 out of 155 = 9.0%), the cut-off amount of cumulative variance was very different across articles (63–98%, Table 1). Similarly, the “greater-than-one” eigenvalue criterion, which was used in 11 articles (7.1%), sometimes results in the inclusion of non-significant axes, especially for data sets with many variables (Vieira, 2012). Similarly, the scree plot criterion, which tends to perform poorly with an increasing number of variables (Vieira, 2012), was used in one article only (0.7%, Table 1).

Table 1 Review of articles published in PeerJ journals that used PCA.

Journal	Criteria for PC selection	Total	
Section	None	Cum.var.	Eigen. > 1	Scree plot		
PeerJ Life & Environment	117	12	11	1	141	
Aquatic Biology	11	0	0	0	11	
Biochemistry, Biophysics & Molecular Biology	10	1	0	0	11	
Biodiversity & Conservation	4	1	2	0	7	
Bioinformatics & Genomics	24	1	0	1	26	
Brain, Cognition & Mental Health	1	1	1	0	3	
Ecology	7	2	0	0	9	
Environmental Science	3	0	1	0	4	
Microbiology	12	2	1	0	15	
Paleontology & Evolutionary Science	3	2	0	0	5	
Plant Biology	28	0	4	0	32	
Zoological Science	11	2	1	0	14	
None assigned	3	0	1	0	4	
PeerJ Computer Science	12	2	0	0	14	
Total	129	14	11	1	155	
Notes:

Results of the review of 155 articles published in PeerJ journals during 2021 (until Nov 30th) that used PCA. The reference list for this review is available in the Table S1.

Cum. var., cumulative variance; eigen., eigenvalue.

The performance evaluations of Vieira (2012) indicated that the chosen statistics used in PCAtest had a good overall performance even in the case of “noisy” data sets containing variables with low correlation or representing linear combinations of the other variables. In particular, the ψ and φ statistics performed very well even in the case of data sets with more variables than observations. The preliminary evaluation of PCAtest with simulated datasets suggests a high accuracy in detecting true correlational structure in the data, but a 5% error when rejecting the null hypothesis of no correlation structure (Type I error). Further exploration of parameter space is necessary to evaluate the conditions (number of variables and observations, correlation patterns among variables) under which PCAtest finds a significant PCA when there is no correlational structure in the data. For the selection of PC axes, PCAtest implements the rank-of-roots statistic, which always had better performance than alternative statistics (Vieira, 2012). However, for the selection of relevant variables, the index of the loadings had a high false positive rate in large data sets (i.e., selecting too many contributing variables with non-significant loadings). Therefore, an option was included in PCAtest to calculate the correlations of the PCs with the variables, in addition to the index of the loadings, for comparison of both statistics. In addition, the problem of ‘axis reflection’ (i.e., the arbitrary permutation of signs among loadings and PC scores), which is well known in the literature (Jackson, 1995; Mehlman, Shepherd & Kelt, 1995; Peres-Neto, Jackson & Somers, 2003, 2005), is effectively avoided with these two statistics as originally implemented by Vieira (2012) and in the R package introduced here.

Traditional parametric, statistical approaches have been used to evaluate significance of a PCA, which assume multivariate normality distribution of the data (Peres-Neto, Jackson & Somers, 2005). Alternatively, non-parametric approaches such as the bootstrap have been used to compare the confidence intervals of PCA statistics with their expected values based on ad criteria as explained above (Peres-Neto, Jackson & Somers, 2003). However, bootstrap techniques do not build an appropriate null hypothesis because the resampling does not remove the correlation structure among the original variables and do not account for axis reflection and reordering (Vieira, 2012). In addition, cross-validation computational approaches have been frequently used for PC selection, but they are considered to be more appropriate when the goal of the PCA is data prediction instead of data exploration (Vitale et al., 2017). On the contrary, permutation approaches that shuffle observations within variables, as implemented in PCAtest, effectively creates an expected distribution of several statistics under the assumption of non-correlated variables which can be used to evaluate the significance of a PCA, to retain the number of relevant PC axes, and the identify the significantly contributing variables to those axes. Vitale et al. (2017) introduced a novel permutational approach that uses the residual matrix for extracting PCs (except for PC1), which increases the sensitivity in detecting relevant PC accounting for small portions of variance. However, they did not provide a software implementation for this alternative algorithm and the statistic they used (an F-like ratio-of-roots) showed high Type I error for trivial PCs in simulations (Vieira, 2012). More recently, Dobriban (2020) rigorously formalized the theoretical foundation of permutation-based, parallel analysis for selecting the number of relevant PCs, and warned about the ‘shadowing’ effect of major PCs with large eigenvalues that could obscure the signal of subsequent, weaker PCs. However, parallel analysis works well, especially in high-dimensional data sets, when a PCA is able to extract PCs with loadings from several observed variables (Dobriban, 2020).

Conclusions

PCA is one of the most popular statistical approaches across many disciplines for data ordination and dimensionality reduction of multivariate data. However, PCA is meaningful and should only be applied when there is a significant correlation structure among the observed variables. Due to sampling variance alone, it is possible that some level of random correlation structure could occur in the data. Although it is necessary to evaluate if the observed correlation structure is higher than expected due to chance alone, this evaluation is rarely undertaken in empirical studies. The R package PCAtest uses random permutation to assess PCA significance and to select the number of significant PCs. In addition, PCAtest can also test the contribution of the observed variables to each significant PC. Therefore, I encourage R users to routinely apply PCAtest to test the significance of their PCA analyses before proceeding with the direct interpretation of PC axes and/or the utilization of PC scores in subsequent evolutionary and ecological analyses.

Supplemental Information

Supplemental Information 1 Results of PCAtest analysis of the example dataset with no correlation among variables.

Observed Psi and Phi values, and eigenvalues obtained from the PCAtest analysis of 100 replicate datasets generated without correlation among five variables and 100 observations. Significant values based on permutation test are indicated with asterisks (P < 0.05). % = percentage of explained variance accounted for the first PC axis.

Click here for additional data file.

Supplemental Information 2 Results of PCAtest analysis of the example dataset with correlation among variables (r = 0.25).

Observed Psi and Phi values, and eigenvalues obtained from the PCAtest analysis of 100 replicate datasets generated with correlation (r = 0.25) among five variables and 100 observations. Significant values based on permutation test are indicated with asterisks (* = P < 0.05, ** = P < 0.01). % = percentage of explained variance accounted for the significant PCs only.

Click here for additional data file.

Supplemental Information 3 Results of PCAtest analysis of the example dataset with correlation among variables (r = 0.5).

Observed Psi and Phi values, and eigenvalues obtained from the PCAtest analysis of 100 replicate datasets generated with correlation (r = 0.5) among five variables and 100 observations. Significant values based on permutation test are indicated with asterisks (** = P < 0.01). % = percentage of explained variance accounted for PC1.

Click here for additional data file.

Supplemental Information 4 Results of PCAtest analysis with the example dataset (seven variables, 29 observations) from Wong & Carmona (2021).

Observed eigenvalues and index of the loadings obtained from the PCAtest analysis of an example dataset containing seven variables and 29 observations (Wong & Carmona, 2021). Significant values based on permutation test are indicated with asterisks (* = P < 0.05, ** = P < 0.01). % = percentage of explained variance.

Click here for additional data file.

Supplemental Information 5 Results of Q-mode PCAtest analysis of the transposed Ringnér (2008)’s microarray data.

Observed eigenvalues and index of the loadings obtained from the Q-mode PCAtest analysis of transposed Ringnér (2008)’s microarray data (8,534 variables, 105 observations). Significant values based on permutation test are indicated with asterisks (* = P < 0.05, ** = P < 0.01). % = percentage of explained variance.

Click here for additional data file.

Supplemental Information 6 Reference list of articles published during 2021 (until Nov. 30th) in PeerJ journals that used PCA.

The literature search used the keyword ‘PCA’ and found 193 articles in total, but 38 of them did not actually apply a PCA analysis. A summary of the results is shown in Table 1.

Click here for additional data file.

Supplemental Information 7 Microarray dataset containing the expression profiles of 8,534 genes screened in 105 samples (Ringnér, 2008).

Click here for additional data file.

I thank the comments and suggestions of two anonymous reviewers that greatly improved the manuscript and the guidance of the Academic Editor for the preparation of the R package.

Additional Information and Declarations

Competing Interests

Author Contributions

Data Availability

The author declared that they have no competing interests.

Arley Camargo conceived and designed the experiments, performed the experiments, analyzed the data, prepared figures and/or tables, authored or reviewed drafts of the paper, and approved the final draft.

The following information was supplied regarding data availability:

The R package PCAtest is available at GitHub: https://github.com/arleyc/PCAtest.

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
