# Peer review of "PCAtest: testing the statistical significance of Principal Component Analysis in R"

_PeerJ, doi:10.7717/peerj.12967_

## Round 0.1 · original submission · Major Revisions

Let me first say that I think your contribution is interesting and potentially quite valuable. However, I am returning the manuscript to you without review because I feel there is a major issue that needs to be addressed before we can proceed further.

Your paper provides an R script to perform previously proposed statistical analyses. As such, it is a software paper rather than a paper presenting conceptually novel ideas. And for a software paper, I would expect higher standards of software engineering than your work currently adheres to.

Specifically, I would like to see:
- A proper R package that can be installed like any other R package. It doesn't have to be on CRAN, but it should meet all major CRAN requirements.
- Documentation that is integrated into the R package, following standard R conventions.
- One or more package vignettes describing provided features etc.
- A suite of regression tests that verify that all major features of the package work, ideally connected to github actions so you verify with every commit to your repository that the package still works.
- Ideally, a package website. This is trivially easy to build with the pkgdown package and github pages.

None of this is particularly difficult or time-consuming. In fact, it should take you at most a few days to make these changes. I'm going to provide you with some resources to help you find your way:
- Book on writing R packages: https://r-pkgs.org/
- usethis package which makes package development extremely easy: https://usethis.r-lib.org/
- Github actions for R, quickstart: https://github.com/r-lib/actions/tree/master/examples#quickstart-ci-workflow
- pkgdown package: https://pkgdown.r-lib.org/

---

## Round 0.2 · Major Revisions

Both reviewers have extensive criticism and concern, and I generally agree with the reviewers' arguments. Please prepare a substantially revised manuscript that addresses all the points the reviewers raise.

Reviewer 1 ·

Basic reporting

This work addresses a fundamental aspect of Principal Components Analysis (PCA) application, that unfortunately has been neglected by the general biological/ecological research community: Raw PCA is only a numerical method (and tool), not comprising statistical methods to estimate the significance of the results obtained, namely, the significant principal components (pc) extracted and the variables significantly contributing to each pc. To obtain such answers, suited statistical methods need to be coupled to PCA. The ecological research community has recursively failed on PCA application, thus presenting results and conclusions poorly grounded, to say the least. The example here presented with the re-analysis of the work by Wong and Carmona (2021) is clarifying and I praise the author for it. In the Discussion, the author gives a remarkable example of how many authors have applied incorrect PCA in articles published during 2021 in PeerJ. This example could already be shown early in the Introduction, regardless of being mentioned again in the discussion.

This problem, of enormous magnitude in ecology, nevertheless occurs also in other fields of science. Therefore, I question the author's choice of restricting its scope (writing and examples) to ecology. I suggest for the author to broaden the manuscript to other fields.

The Introduction presents the problem well. The Discussion and conclusion expand on it and present the solution very well. The proposed solution is well grounded on the results. The author did a very good job.

The English, text structure and figures comply with the journal’s requirement.

The presentation of the software, algorithm, processes and artificial data generation is vague. Reproducibility is not possible from the text alone. The article is not self-contained and, for the reader to fully understand it, in parallel he must consult the key references as well as the R script. Personally, I do not find this the best writing style and I prefer a more complete version of the manuscript. Nevertheless, I understand the intension to provide a lighter text enabling a smoother reading flow. But, in such case, the text can be improved and, at the proper places, it must be mentioned (the citations, the R script) where the respective theory/method is thoroughly detailed.

The citations are inadmissibly poor and biased, particularly in the introduction, neglecting key authors in the topic and sometimes incorrectly attributing credit to the wrong author. I also notice that the author never cites more than one reference per idea/message (set of sentences), which is utterly poor and unfair to previous authors. PCA was originally developed by Pearson (1901) and later by Hoteling (in the 1930’s). This citation is mandatory. Other authors as Ricker, Manly, Jackson or Jolliffe are classical references for PCA.
Pearson, K. (1901) On lines and planes of closest fit to systems of points in space. Philos. Mag. 2:559-572.
Manly, B.J.F. (1986) Multivariate Methods. Chapman & Hall, London, UK
Manly, B.J.F. (1991) Randomization and Monte Carlo Methods in Biology. Chapman & Hall, London, UK
Jackson, J.E. (1991) A User’s Guide to Principal Components. John Wiley & Sons, New York, USA
Jolliffe, I.T. (2002) Principal Component Analysis. Springer-Verlag, New York, USA
Jolliffe, I.T. and Cadima, G. (2016). Principal Components Analysis: a review and recent developments. Phil. Trans. Rl Soc. A. 374: 20150202.

All these would be better references for the introductory sentence in lines 27-29. They are particularly better suited than Dobriban (2020), which is not even a good citation for PCA per se but only for the application of permutation tests to PCA and FA. Following the author’s rationale, Vieira (2012) or Björklund (2019) would be as good citations for lines 27-29 as Dobriban (2020) or Abdi and Williams (2010). In fact, Vieira (2012) is a citation suited for most of the sentences in the introduction, as are many of the suggested below. Although the works by Vieira (2012), Vitale (2017), Björklund (2019) and Dobriban (2020) are the recent trend in the application of Permutation Tests to estimate the significance of results from PCA, they were not the original developers of this topic. In fact, most of their works were compilations - or built on top of – previous works. The application of Randomization methods to multivariate analysis (i.e., Permutation Tests for the estimation of significances, as well as Bootstrap and Jacknife for the estimation of confidence intervals) was started by researchers as Legendre, Peres-Neto, Jolliffe, Lebart, Jackson, Dijkseterhuis, Efron, Mehlman, ter Braak, Zwick, Velicer or Westerhuis. These citations became ‘classical’ and are mandatory to the respective sentences in the introduction. I suggest for the author to, at least, look up in the works by Vieira (2012), Vitale (2017), Björklund (2019) and Dobriban (2020) where the aforementioned references fit in his introduction.

References for the bootstrap method are not provided in the Introduction nor on the Methods, although plenty exist. These are some of the suggested above.

Experimental design

The Research question is well defined, relevant and meaningful. It is stated how the research fills an identified knowledge gap.

The research is within Scope of the journal. However, it is subjective whether it can be considered “Original primary research”. The novelty in this work is that it transposes to R software the Matlab software developed by Vieira (2012), updated with a posterior algorithm developed by Björklund (2019) and with bootstrap methods for the estimation of confidence intervals (references missing). Personally, I find this very useful to the scientific community, and worth publishing. Regarding this aspect, the author provided a very good and convincing presentation.

About “Methods described with sufficient detail and information to replicate:
Testing the method with artificial data, thus proving its validity as well as advantages, is a standard in the presentation of new numerical/statistical methods, which the author wisely followed. However, its implementation was poor:
1. How was the artificial data generated? I would prefer to see this better explained in the article. How was the algorithm for the generation of the random data? And the range of coefficients tested? What were the variances of the variables? Their associated errors (ε)? Their distributions? Their imposed effective correlations or covariances? And how? Slopes of regression lines centred on the means? Other? A few added lines suffice to satisfy this requirement.
2. The magnitude of the conditions covered by the artificially generated data seems week. In his presentation, the author should broaden the coverage of the aspects mentioned in the point 1 above. Otherwise, the author explicitly manifests that this is just an example to demonstrate the argumentation, and that its generalized prove was already done somewhere else (cite the authors who used permutations tests in PCAs applied to artificial data).

About “Rigorous investigation performed to a high technical & ethical standard”, some improvements should be done:
Relative to the PCA applied to the Ringnér (2008) data set, my personal experience is that PCAs applied to data sets with so many variables fail completely, particularly if, on the other hand, the number of observations is so small. Ringnér (2008) and the current author, have two options: (1) transpose the data matrix and do the PCA in Q-mode. (2) use the data matrix “as is” to do a PCA in R-mode, but doing it in several steps to iteratively eliminate irrelevant variables (uncorrelated with any other), thus reducing the size of the data set. The irrelevant variables bring noise that veils the true correlations. As this noise is step-wise eliminated, the true correlations emerge. A PCA as allegedly done by Ringnér (2008) is ridiculous.

The PCA applied to the Ringner (2008) data set only used 100 random simulations, which is too little. I usually use 10000. If the data set is very large, bringing computational constrains, 1000 would be admissible. 100 is inadmissible. I highlight the following aspect: with 100 replications, each represents a jump of 1% in p, meaning that p is represented at a 1% resolution. Therefore, around the α= 5% threshold, p jumps from 4% to 5% to 6%. With 1000 replications, each represents a jump of 0.1% in p. With 10000 replications, each represents a jump of 0.01% in p. Therefore, around the α=5% threshold, p jumps from 4.99% to 5% to 5.01%.

Validity of the findings

The Introduction presents the problem well. The Discussion and conclusion expand on it and present the solution, very well. The proposed solution is well grounded on the results. Replication is possible using the R script and the data is provided. Overall, the author provides a very convincing case.

Additional comments

Lines 30-31: “… the first component (PC1) accounts for most of the variance in the data …”. The use of “most” implies that pc1 accounts for more variance than all the other pcs pooled together, which is false. In an hypothetical example, accounting for 40% of the variance, pc1 may account for the largest chunk of variation, but it is certainly not the most as 60% are still unaccounted for.

Line 32: pc scores are usually called z-scores, but can also be called l-scores, depending on the use of covariance or correlation matrices and the standardization method (see Jackson, 1991). At least mention the z-score nomenclature.

Line 34: … can be used for bi- or tri-ordination plots. I personally dislike this solution as, to the unspecialized reader, it suggests some sort of correlation between successive pcs. This is exactly the opposite of what PCA does, extracting pcs subject to the restriction of being orthogonal (or orthonormal if the correlation matrix is used). (see suggested literature on PCA method).

Line 35-37: Very true. And much better alternative to Canonical Correlation Analysis, since there is nothing in nature compelling pc1 from the abiotic variables to be better correlated with pc1 of the biotic variables, pc2 from the abiotic variables to be better correlated with pc2 of the biotic variables, and so forth, as was well presented by Manly (1986) and by Jackson (1991), and well demonstrated by Cabaço et al. (2008) on the analysis of Zostera marina in Ria de Faro.

Line 41-42: This demonstration rouse way before Björklund (2019). At least by Stauffer et al (1985), Jackson (1993), Dijkseterhuis and Heiser (1995), Peres-Neto et al (2005), Vieira (2012), Jolliffe and Cadima (2016) and Vitale (2017). Hence, this sentence written “as is” and only with this citation, is false and inadmissible. The previous authors deserve the due credit.

This problem repeats throughout the manuscript. I will restrain from correcting every single incorrect citation. I leave that task for the author; who must revise extensively the citations and the criteria used to chose them.

Lines 42-45: This is a very important aspect to analyse the data from Ringnér (2008) and discuss it. But it belongs there, in the results and discussion. In the introduction, it is a bad example since the researcher has always the option to transpose the data matrix and choose between an R-mode (focus on the variables) or Q-mode PCA (focus on the observations) (see for example Lee et al. 2017). In such case, the observations act as variables and each pc weights the observations. Z-scores are estimated for each variable. The results from R-mode and Q-mode PCA applied to the same data set come hand-in-hand and can be easily transposed. Furthermore, see argumentation by Jolliffe and Cadima (2016). The fundamental aspect of these lines should be that, even if all variables are uncorrelated, just due to random error, larger and smaller pcs will inevitably arise, implying pcs higher than 1 that still only report to random associations.

Lines 72-73: This is false in the sense that the permutation procedure was already proposed by Vieira (2012), and by others before him (see section above).

Line 73: “Vieira”

Line 99: this statistic was developed by Vieira (2012) and not by Björklund (2019).

Line 105-106: True, but still a poor and partial explanation. The fundamental aspect is that significant pcs represent factual associations between variables, whereas non-significant pcs represent only random associations.

Line 106-107: good choice. But it should be explained why: an α level is only a probability of being wright or wrong when making a decision of accepting or rejecting a null hypothesis. Hence, p<0.05 is not a warranty of rejection nor is p>0.05 a warranty of its acceptance. Statistical methods are tools helping the decision-maker, and not dogmas around 0.05 α levels. The last call should always belong to the human in charge, and the impact of making that call must always be crucial to the decision-making process.

Line 146-154: More information about the structure of the artificial data should be presented. Was the PCA done over the correlation matrix or the covariance matrix?

Line 191-192: This is the first, fundamental sentence setting the tone to the discussion. Yet, it completely misses the global picture by focusing exclusively on the topic of Life Evolution. Incorrect PCA application is spread all over sciences, including all topics covered by PeerJ.

Reviewer 2 ·

Basic reporting

There is a conflation of permutation tests and bootstrap throughout the article. A permutation test involves a choice of a null hypothesis, which is unclear throughout the manuscript; the author simply refers to random noise without giving it a distribution or probability. A bootstrap, by contrast, is a means for quantifying the uncertainty in the parameter. The author needs to define these terms and goals much more carefully in the paper.

Experimental design

The author evaluated the bootstrapped PCA procedure and permutation PCA procedure on simulated and real datasets of varying complexity.

1. For the simulated datasets, the authors needs to make clear what true model is from a latent factor perspective. I think that the covariance matrix S with exchangeable correlation is equivalent to a

2. Concluding something from one simulated dataset about the frequentist properties of a procedure is not sufficient. The author needs to simulate datasets a certain number of times and average the estimators over the simulations.

3. There is a sign issue with PCA in that one can multiply all components of the PC's by negative one and still explain the same amount of variation. This issue is not addressed.

4. It is unclear what knowledge gap the article addresses.

Validity of the findings

1. Given the simulated data, the author needs to perform repeated simulation data generation rather than a single dataset generation.

2. R code is available to reproduce the findings.

3. The conclusion that there needs to be better PCA analyses is an appropriate one. However, the literature search the author was not systematic.

---

## Round 0.3 · Minor Revisions

Thank you for your careful revisions. Reviewer 1 has a few more comments that shouldn't be difficult to address.

Reviewer 1 ·

Basic reporting

The manuscript is much better now, and I congratulate the author for it. I can only wish that finally ecologists start being more careful with their applications of PCA. This manuscript can certainly give a contribution to it.
It is almost ready for publication. But I still see a few problems. Some sentences in the introduction are too long, becoming hard to follow and leading to confusion. Lines 67-71 or 80-85 are good examples. I suggest for the author to, generally speaking, try to use shorter sentences.
Some incorrect citations, persist. I will give the examples. In some cases, they are simply incorrect. In other cases, I personally would give more than one reference i.e., at least the original together with some very commonly red and cited.
Line 30: Before Cadima and Jollife (2016), the same had already been showed at least by Manly (1986), Jackson (1991) and Vieira (2012).
Line 38: The orthogonality of pc axis had already been explained at least by Manly (1986) and Jackson (1991).
Line 40-42: deserves references as Manly (1986), Jackson (1991) and Vieira (2012).
Lines 44 to 69: Manly (1986) and Vieira (2012) are actually better references than the ones used. Manly (1986) debated this earlier while Vieira (2012) debated this deeper.
Lines 55 to 57 about the SCREE Plot are repeated in lines 62 to 65. Or is there a small difference between them? A comparison against randomly generated data, in the second case? But then, separating both cases with other text in-between might not be a good idea. Vieira (2012 - Fig 2) demonstrated this SCREE Plot before Bro and Smilde (2014), having taken it from Mainly (1986) and (Jackson (1991).
Lines 58-59: Again, Manly (1986), Jackson (1991) and Vieira (2012), among many others, already debated this before Bjorklund (2019).
Lines 58-59: Besides Ringer (2008), Manly (1986), Jackson (1991) and Vieira (2012), among many others, also debated this. With a big difference that Ringer (2008) only applied to a data set whereas Manly (1986), Jackson (1991) and Vieira (2012) debated about the validity of its application, debates that were inserted in whole articles or books about the PCA method.
Lines 67-71: the author mixes in the same sentence two entirely different issues, generating a sentence that is incorrect:
i) ad-hoc tests rely on false assumptions and generally provide incorrect results leading to false conclusions. Thus, randomization tests are better suited.
ii) Among randomization tests, there are two classes: Permutation tests propose a null hypothesis and estimate significances by randomly redistributing observations breaking correlations among variables. Bootstrap and Jacknife estimate confidence intervals by randomly resampling observations without breaking correlations among variables
Consequently, the claim “Because these ad hoc procedures …Vitale et al. (2017) suggested that permutation-based tests are better suited than other randomization approaches …” is incorrect.
Or then, it is me not quite understanding the text. The author mentions the ad hoc tests and statistical procedures. But the methods described above are generally not “statistical”.
Line 80-85: explain the concept in several, shorter sentences.
Lines 87-89: True that the metric  was proposed by Gleason & Staelin (1975) whereas  was proposed by (Vieira, 2012). However, both where proposed to be applied with permutation tests, along with other 5 metrics, by Vieira (2012) i.e., before Bjorklund (2019).

Overall, I strongly encourage the author to read carefully Vieira (2012) in order to be better informed about who did what. In such case, the author will find information that is very helpful for the problem about “axis reflection”:
Axis reflexion i.e., the permutation of signs among loadings and eigenvalues, and the implications it brings for their ranking in randomization methods, was first debated by Jackson (1995), Mehlman et al. (1995) and Peres-Neto et al. (2003 and 2005). This problem was solved by Vieira (2012) in two ways: (i) when using the “index of the loadings”, both the loadings and the eigenvalue are squared. Therefore, their original sign is irrelevant for the subsequent ranking. (ii) when using the “correlation coefficient” between pc and variable, Vieira (2012) tested ranking according to the absolute value.

Given that the present R-software is an adaptation from the Matlab software by Vieira (2012), this R-software should also be robust against axis reflection.

Lines 96-101: I may be wrong, but my feeling is that the major difference between Matlab and R is that the first is paid whereas the second is free-ware.

Line 162: “Index of the loadings”, right?

The text in lines 177 to 192, relating to Figs 1-3, should also have a table showing the eigenvalues, and loadings of each pc, and the significant ones highlighted (in bold or in color). This would help the reader understand the real example demonstrating the theory.
The above applies also to the analysis of the real data sets.

The figure from the analysis run in Q-mode should also be in the main text, and not supplementary material, and striking better that, in cases like this, Q-mode should be preferred to R-mode.

The analysis and the examples are very good. Bulls-eye! The Results and Discussion are very well written. I would not do better! My congratulations to the author.

Raw data and software are shared. The article is self-contained with relevant results and hypothesis.

Best wishes

Experimental design

Weaknesses from former submission were solved. Currently, all checks.

Validity of the findings

Weaknesses from former submission were solved. Currently, all checks.

Reviewer 2 ·

Basic reporting

The author presents much improved work in the revision. I have nothing to add.

Experimental design

The author presents much improved work in the revision. I have nothing to add.

Validity of the findings

The author presents much improved work in the revision. I have nothing to add.

---

## Round 0.4 · accepted · Accept

Thank you for your last round of careful revisions.